# Treatment of Periprosthetic Joint Infection with Intravenous Vancomycin: Do We Hit the Target?

**DOI:** 10.3390/antibiotics13121226

**Published:** 2024-12-18

**Authors:** Rasmus Haglund, Ulrika Tornberg, Ann-Charlotte Claesson, Eva Freyhult, Nils P. Hailer

**Affiliations:** 1Orthopaedics/Department of Surgical Sciences, Uppsala University, SE-751 85 Uppsala, Sweden; ulrika.tornberg@akademiska.se (U.T.); ann-charlotte.claesson@uu.se (A.-C.C.); nils.hailer@uu.se (N.P.H.); 2Science for Life Laboratory, Department of Cell and Molecular Biology, National Bioinformatics Infrastructure Sweden, Uppsala University, SE-752 37 Uppsala, Sweden; eva.freyhult@icm.uu.se

**Keywords:** periprosthetic joint infection, vancomycin, trough value, drug monitoring, therapeutic range, antibiotic-loaded cement

## Abstract

**Background/objectives**: Vancomycin is commonly used in the treatment of periprosthetic joint infection (PJI), and trough concentrations are measured to ascertain that they are within the therapeutic range. It has not been investigated what proportion of vancomycin concentrations during treatment of PJI patients is accurately within this range, how many dose adjustments are commonly needed, and which patient factors predispose towards aberrations from the desired range. **Method**: In this single-center cohort study, we investigated vancomycin trough concentrations in 108 patients with surgically treated PJI who received IV administered vancomycin treatment post-operatively. Patients were identified in our local arthroplasty register, and data beyond what was available in the register were collected from electronic medical charts. **Results**: Of the final study cohort, 41% were women, and the median age was 71 (IQR 63–79) years. Most patients had PJI of the hip (73%), the majority (54%) underwent a debridement, antibiotics and implant retention (DAIR) procedure prior to vancomycin treatment, and 39% received vancomycin-loaded bone cement during the preceding revision procedure. Of 791 vancomycin trough measurements, only 58.2% were within the target range of 15–20 mg/L, 18.5% were below, and 23.4% were above. A total of 71% of all patients required at least one dose adjustment, and the median length of vancomycin treatment was 8 days. We observed positive correlations of vancomycin trough concentrations with both age (Spearman’s rho = 0.35, *p* < 0.001) and pre-treatment creatinine concentrations (Spearman’s rho = 0.34, *p* < 0.001), but no statistically significant difference between patients who had received vancomycin-loaded bone cement and those who had not. **Conclusions**: In our PJI patients, a high proportion of vancomycin trough concentrations were outside the therapeutic range, despite adherence to local and national guidelines. We can also confirm that caution needs to be exerted in patients of advanced age and those with compromised kidney function. Alternative broad-spectrum antibiotics that do not require as extensive therapeutic drug monitoring should be further explored.

## 1. Introduction

Periprosthetic joint infection (PJI) is a serious complication following arthroplasty surgery. The incidence is between 1% and 2% for hip or knee PJI, some studies suggest that the actual incidence of PJI may be underestimated, and the incidence is increasing, at least in the Nordic countries [1,2,3,4]. PJI develops through adherence of microorganisms to the implants’ surface which can occur through different mechanisms. Hematogenous spread, perioperative inoculation or postoperative contamination, most often in the context of a surgical site infection, are the common underlying infection pathways [5]. Several definitions of PJI have been proposed, with the most recent and already widely accepted being that of the European Bone and Joint Infection Society (EBJIS) [6]. This definition considers clinical features, clinical chemistry, microbiology, histology and radiology, and defines three categories: (1) “infection unlikely”, which means there are no signs of ongoing infection, (2) “infection likely”, which means there are at least two findings that indicate that there are an ongoing infection, and, (3) “infection confirmed”, which means there are at least one sign which confirms there are an ongoing infection [7]. Based on the time of onset of symptoms, PJI can also be classified into three different categories. Firstly, early postoperative PJI occurs within one month postoperatively and is typically contiguous in nature. Secondly, acute hematogenous PJI arises after an uneventful postoperative period, with bacteria spreading through the bloodstream. Finally, chronic PJI manifests after the early post-interventional phase, persisting for more than three weeks. Both hematogenous and contiguous mechanisms of microorganism spread can be implicated in the development of chronic PJI [8].

The treatment of PJI involves a combination of surgical and antibiotic approaches. Thorough debridement with retained implants (“debridement, antibiotics and implant retention”, DAIR) and one- or two-stage implant exchanges are the principal treatment options [8]. DAIR procedures can be successfully performed in acute hematogenous PJI of short duration and in early postoperative PJI that manifests within one month after index surgery. In contrast, chronic PJI generally requires implants to be removed within the setting of either one- or two-stage exchanges. The choice of antibiotics is guided by tissue culture results, classification of the infection, the type of surgery that has been performed, and, in the absence of conclusive cultures, clinical suspicion. Both after DAIR procedures and after one- or two-stage exchanges antibiotics with a relatively broad antimicrobial spectrum are empirically administered prior to confirmation of causative agents by tissue culture, implant sonication, or molecular study. Due to its antimicrobial spectrum, vancomycin is commonly used at this stage of PJI treatments.

Vancomycin is a tricyclic glycopeptide antibiotic that acts by inhibiting cell wall synthesis in Gram-positive bacteria. Vancomycin binds to the D-alanyl-D-alanine terminus of the cell wall precursor unit, thus preventing the polymerization of pentapeptide and N-acetylglucosamine into peptidoglycan [9,10]. Vancomycin is active against staphylococci, including methicillin-resistant *S. aureus* (MRSA), coagulase-negative staphylococci (CoNS), and enterococci. Since vancomycin is poorly absorbed in the gastrointestinal tract it is intravenously (IV) administered [5]. The serum protein binding of vancomycin is around 55%, its half-life is between 4 and 6 h, and a steady-state concentration is usually reached during the second day of treatment. Vancomycin has no known active metabolites and is primarily eliminated renally by glomerular filtration. Measurement of trough levels immediately prior to the next intended dose is performed to monitor serum concentrations, aiming for a target range of 15–20 mg/L in our local guidelines that are based on national recommendations of the Swedish Society of Infectious Diseases [5]. Given normal renal function, patients usually receive a loading dose of 30 mg/kg vancomycin (in clinical practice often 2 g), followed by 1 g every 8 h. Adjustments are made based on measured trough values, and both dosage intervals and doses may be changed to achieve a steady state within the above-mentioned concentration range [10].

In addition to systemic vancomycin treatment, the use of antibiotic-loaded bone cement (ALBC) containing vancomycin is common when the index PJI revision procedure is a one- or two-stage exchange. In one-stage exchanges and in the second stage of two-stage exchanges, the definitive implants are often, but not always, fixed using such vancomycin-containing ALBC, and in the first stage of two-stage procedures, articulating or non-articulating spacers are very often manufactured from vancomycin-containing ALBC. This approach renders a dual exposure of PJI patients to vancomycin, adding local vancomycin release to the systemic administration.

PJI patients are mostly elderly and often quite severely co-morbid, and there is limited research on the accuracy of vancomycin treatment in terms of the proportion of vancomycin concentrations within the therapeutic target range in this specific group of patients [11]. The effects of combining vancomycin-loaded bone cement during revision surgery followed by IV administration of vancomycin on systemic trough concentrations have also not been explored in PJI patients. The primary aims of this study were thus (1) to investigate whether obtained serum vancomycin concentrations in patients with PJI are within the desired concentration range, and (2) to explore how many trough value measurements and dose adjustments are necessary during a standard treatment regimen. The secondary aims were to investigate potential correlations of vancomycin trough concentrations with serum creatinine levels, body mass index (BMI), age, and sex, and, finally, to assess whether additional exposure to vancomycin-loaded bone cement in PJI patients treated with IV vancomycin affects serum vancomycin concentrations.

## 2. Results

### 2.1. Description of the Study Population

The final cohort included 108 patients with a median age of 71 years and a median BMI of 27.4 (Table 1). Most PJI involved hip prostheses (73%), and the most common revision surgery preceding vancomycin treatment was a DAIR (54%). A total of 42 (39%) patients received ALBC with vancomycin (Copal G+V^®^), the remaining 61% received no cement at all, either because they underwent a DAIR procedure during which only mobile components were exchanged, or because they received uncemented implants during revision surgery. In total, 22 patients were discharged with ongoing vancomycin treatment to their referring hospitals. Our patients had a median of 8 (range 1–24) days of IV vancomycin treatment. An analysis of tissue culture results was performed, where the most commonly occurring causative organism was *S. epidermidis* (see Appendix A).

### 2.2. Trough Concentrations

A total of 791 trough concentrations of vancomycin measurements were performed, with 58.2% within the target range, 18.5% below, and 23.4% above (Figure 1 and Figure 2). In total, 69% of patients had at least one trough concentration above the desired concentration range, and 66% had at least one below. Higher age was associated with a higher percentage of trough measurements above the target range (rho = 0.35; *p* < 0.001; Figure 3). Higher serum creatinine levels on the first day of treatment correlated with higher trough concentrations (rho = 0.34; *p* < 0.001; Figure 4). There was no statistically significant correlation of BMI with trough concentration levels outside the target range, and we also found no statistically significant difference in initial trough concentration of vancomycin between patients who received ALBC with vancomycin and those who did not (*p* = 0.84). The average vancomycin trough value for the group of patients who did receive ALBC was 16.39 (SD = 0.19), and it was 16.12 (SD = 0.19) for those who did not.

### 2.3. Dose Changes

The administered vancomycin doses required frequent changes, and in only 29% of the study population were no changes made from the standard regimen (Table 2). The average number of dose adjustments was 1.2 per patient.

### 2.4. Renal Function

Renal function was consequently followed in all patients by repeated measurements of creatinine concentrations. We observed elevated creatinine concentrations above 150 µmol/L mostly during the first six days of vancomycin treatment, whereas very few similarly elevated vancomycin concentrations were found after day seven (Figure 5). Due to the limited number of values exceeding the reference range, a formal regression analysis investigating risk factors underlying elevated creatine concentrations was not feasible.

### 2.5. Treatment Costs

We estimated the costs of vancomycin application in our cohort of PJI patients. For our healthcare region, vials with dry substance contain 0.5 g and 1 g of vancomycin cost SEK 66 or SEK 88, respectively, excluding dispensing fees. The total cost of one vancomycin dose ready to administer (RTA) amounts to SEK 220. Each trough concentration analysis costs SEK 94. Additional costs, such as manual labor, the use of protective equipment, additional supplies, and intravenous access, were not included. Building on this simple calculation, an 8-day hospital stay with 1 g vancomycin administered three times daily would cost SEK 5282, with monitoring adding SEK 754. Including our average cost per in-patient day, this amounts to a total cost of SEK 86,684, or approximately USD 8000 [12].

## 3. Discussion

Vancomycin is commonly used in the treatment of PJI due to its efficacy against Gram-positive bacteria, including MRSA. In PJI, vancomycin is almost exclusively administered IV, but local delivery methods, such as vancomycin-loaded bone cement, are sometimes additionally employed to achieve high local tissue concentrations. In our study, vancomycin trough concentrations were unexpectedly often outside the therapeutic target range, and dose adjustments were necessary in the vast majority of patients, with about 10% of our cohort needing as many as three or more adjustments. Patients with higher age and those with reduced eGFR were more likely to reach supratherapeutic vancomycin concentrations. In contrast, patients who were simultaneously exposed to both local and systemic vancomycin administration did not have higher serum vancomycin concentrations than those who were only treated systemically.

### 3.1. Limitations and Strengths

Due to our retrospective study design, which relies on the review of electronic medical charts, incomplete or erroneous records and missing information are to be expected. Consequently, roughly a third of all eligible patients had to be excluded due to such errors. Even in the final study population, analysis of certain variables of putative interest was incomplete, such as the investigation of the incidence of common side effects of vancomycin.

Rapid infusion of vancomycin can cause phlebitis, pseudo-allergic reactions, and upper body flushing (“red-person syndrome”). Other common side effects (≥1/100 to <1/10) include hypotension, dyspnea, stridor, pruritus, urticaria, exanthema, mucosal inflammation, and renal insufficiency. A less common side effect (≥1/1000 to <1/100) is temporary or permanent hearing loss [9]. Renal insufficiency is a significant concern with vancomycin, given its renal elimination and its nephrotoxicity. Various studies report inconsistent findings regarding vancomycin-induced nephrotoxicity. Filippone et al. (2017) reported incidences ranging from 0% to over 40%. Risk factors for vancomycin-induced nephrotoxicity include high vancomycin trough concentrations, obesity, pre-existing renal insufficiency, severe illness, and concurrent nephrotoxin use [13]. Horey et al. showed that increasing trough concentrations of vancomycin were directly proportional with higher risk of nephrotoxicity [14]. Studies on the ototoxicity of vancomycin vary. Humphrey et al. (2019) reported an 8% incidence among nearly 100 patients, mostly presenting mild to moderate hearing loss [15]. Forouzesh et al. (2009) found a 12% incidence in their larger study, with all affected patients being over 52 years old, suggesting increasing age as a risk factor underlying this specific complication [16]. However, according to the Swedish Medical Products Agency, ototoxicity is an uncommon side effect of vancomycin treatment (≥1/1000 to <1/100) [9].

Serious or even life-threatening conditions, such as severe hypotension, renal failure or notable hearing loss, would have been noted in our charts, but no such event was recorded. Our analysis of creatinine concentrations over time also suggests that no case of acute renal failure occurred in our cohort. However, the absence of documentation of less serious side effects cannot be taken as evidence that no such side effects occurred, and we therefore chose not to analyze their occurrence.

Due to the fact that our study was solely based on a local registry and local electronic medical charts, another limitation of our study is that we were unable to follow the clinical course of patients referred to us from other hospitals who were discharged to their home hospital with ongoing vancomycin treatment. For this group of patients, we lack information on subsequent trough values and the total duration of treatment, and their observation was therefore censored at the time of transfer to their referring unit.

Being a single-center study restricts the generalizability of our findings. The unique practices, protocols, and resources of our institution have surely influenced the results, limiting their applicability to other settings or healthcare environments. These factors should be considered when interpreting the study’s conclusion.

Our choice to analyze only trough concentrations is open for debate. The systemic application of vancomycin requires intensive monitoring to ensure efficacy while minimizing side effects. The target trough concentration of vancomycin is 15–20 mg/L [17,18]. At our unit, the trough serum vancomycin concentration and glomerular filtration rate (GFR) are measured before every third dose [17]. However, according to Álvarez et al. (2016), the ratio of the area under the curve (AUC) divided by the minimum inhibitory concentration (MIC) is a more accurate tool for monitoring vancomycin [18]. However, using AUC and MIC requires more expertise and time compared to the relatively simple assessment of trough values. Thus, at our unit, the trough value is used as a surrogate measure, although some studies indicate that this may not be sufficient [19,20]. A recent study using a machine learning algorithm by Chen et al. suggests that only the daily vancomycin dose and the GFR are required to accurately predict vancomycin through concentrations, an approach that may facilitate the determination of vancomycin doses, without using the slightly more cumbersome ratio of AUC by MIC [20].

There are some strengths to our study. According to several guidelines and consensus documents, vancomycin is the antibiotic of choice when treating oxacillin-resistant staphylococci or enterococci, and a preferred alternative treatment in streptococcal PJI [21,22]. Our use of vancomycin in the investigated cohort of PJI patients is thus well aligned with nationally and internationally established practice. Our local routines on the monitoring of vancomycin treatment is in agreement with national recommendations [23,24], although recent consensus guidelines from the Infectious Diseases Society of America advocate the use of the AUC/MIC ratio to avoid nephrotoxicity and to achieve clinical efficacy [25]. Our diagnosis of PJI is stringent and was based on the current EBJIS criteria, and all patients in our cohort had PJI confirmed by at least two positive culture specimens obtained during index surgery.

### 3.2. Accord and Discord with Previous Studies

Our finding of a substantial proportion of vancomycin-treated patients receiving supra- or infra-therapeutic concentrations, despite stringent adherence to local and national guidelines on the use and dosage of vancomycin, is by no means unique. The phenomenon has been relatively extensively studied in obese patients, and a systematic review on this topic concluded that despite slightly modified or even extensively altered dosing protocols, considerable numbers of patients have initial trough concentrations outside the intended range [26]. Trough concentrations of vancomycin have only a moderate correlation with the AUC, with an R^2^ of only 0.51 in one study [27]. Nonetheless, measurements of AUC or AUC/MIC have been advocated to render a more precise monitoring of vancomycin treatments [18].

Our finding that age and serum creatinine correlate well with vancomycin trough concentrations is in line with previous findings, and the association of these parameters with each other has been used to develop dosing nomograms that facilitate the selection of appropriate initial vancomycin regimens [28].

We found no local or systemic signs of adverse reactions to locally applied vancomycin-containing ALBC, in line with previous reports [29,30]. In the study by Kendoff et al., 20 patients received the industrially manufactured ALBC Copal G+V^®^ for fixation of hip arthroplasty components inserted during one-stage exchanges for pre-existent PJI [29]. These authors found a maximal vancomycin plasma concentration of 0.74 mcg/mL, thus far below the intended therapeutic range after systemic vancomycin application. Higher concentrations of up to 3 mcg/mL, thus still well below the therapeutic range, were measured in the study by Oe et al. [30]. These reports support our finding that PJI patients treated with IV administered vancomycin are not at danger of developing excessive trough concentrations of vancomycin when receiving additional local vancomycin in the form of ALBC. A formal correlation analysis of local and systemic concentrations of vancomycin after the use of vancomycin-loaded bone cement has, to our knowledge, not been performed. That said, some authors have investigated concentrations of vancomycin both in wound drainage fluid and in blood [29]. However, it has been shown that vancomycin concentrations in samples from tissue surrounding ALBC spacers correlate with the amount of ALBC that was used to mold the spacers [31].

Our study was not primarily intended to answer health economic questions. Nonetheless, when observing the high number of trough value assessments per patient, the relatively frequent dosage adjustments, and the total length of stay related to the IV administration of vancomycin, we found it tempting to address such aspects. Briefly, we found that the cost for an 8-day course with 1 g vancomycin administered three times daily generates costs of around SEK 86,000, including the costs for in-patient treatment. In contrast to vancomycin, dalbavancin is significantly more expensive, with a total cost of SEK 21,564 per dose ready to administer. A study by Rappo et al. showed that two doses of 1500 mg dalbavancin administered on days 1 and 8 was comparable to 4 to 6 weeks of IV vancomycin treatment for osteomyelitis [32]. The use of dalbavancin could potentially eliminate the need for inpatient care solely for antibiotic administration, which is required when administering IV vancomycin. However, it should be noted that the above cited study included only 80 patients, the randomization was conducted at a 7:1 ratio for dalbavancin versus standard of care, and the cohort did not suffer from PJI but from osteomyelitis. Unfortunately, no larger randomized controlled trials investigating dalbavancin have been conducted to date. Nonetheless, dalbavancin has been tentatively used in PJI patients even at our unit, and patients on dalbavancin could thus be discharged once their condition improved after surgery, potentially leading to a shorter hospital stay. Moreover, dalbavancin does not require as extensive therapeutic drug monitoring as vancomycin, such as the repeated measurement of trough levels. The Swedish Society of Infectious diseases recommends therapeutic drug monitoring only when available, and if so, only one blood sample obtained after 3–5 weeks is recommended, depending on eGFR [5]. To summarize, a 3- to 5-day inpatient treatment regimen of a patient with dalbavancin instead of vancomycin would cost between SEK 51,807 and SEK 73,371, excluding therapeutic drug monitoring [12], and would thus potentially render substantial savings and a reduced need for hospitalization.

Another alternative to IV vancomycin treatment in PJI patients is the use of daptomycin. Daptomycin can be administered on an outpatient basis since it requires only once-daily dosing. Although this treatment enables the patient to be discharged from inpatient care, it necessitates substantial planning, at least in the Swedish healthcare setting. Specific agreements with various stakeholders in the municipal home healthcare system to secure their approval for managing administration, blood sampling, and central venous access care have to be made. Additionally, blood concentrations of creatine kinase have to be monitored in order to detect the potential side effect of rhabdomyolysis [5]. Taken together, the use of daptomycin appears slightly more cumbersome from a logistic perspective.

## 4. Patients and Methods

### 4.1. Study Design and Patient Selection

We designed an observational, longitudinal, single-center cohort study including patients treated with vancomycin for hip or knee PJI at a tertiary referral center from 23 November 2017 to 4 August 2023. The starting date was chosen because the medication module in our electronic medical records system underwent a major update in October 2017, allowing for the continuous registration of vancomycin doses and dosage intervals from then on. By searching our local arthroplasty register, we initially identified 158 patients who were surgically treated for PJI and who received IV administered vancomycin treatment for their PJI. Exclusion of those not meeting inclusion criteria, mostly due to having received prior vancomycin regimens, and of those with incorrect or missing records rendered a final population of 108 patients, none of whom was bacteremic after their index procedure (Figure 6).

### 4.2. Antibiotic-Loaded Bone Cement

At our institution, standard Copal G+V^®^, supplied by the manufacturer Heraeus (Wehrheim, Germany), is the most commonly used ALBC. Copal G+V^®^ contains 2 g of vancomycin per 43 g pack of cement powder, in addition to 0.5 g gentamicin. The number of packs used per patient varies depending on the implants used, with one pack for the acetabular component and one pack for a standard primary femoral component being the minimum when using cemented implants. In select patients, however, up to four packs could be used, one for the acetabular component and an additional three for a long-stemmed femoral revision component.

### 4.3. Variables and Data Collection

The baseline data, including age, sex, BMI, date of surgery, affected joint (hip or knee), type of revision surgery, and the use of ALBC at index revision surgery, were obtained from the local arthroplasty register. In addition, the data were collected from electronic medical charts and added to the register, including the start date of the vancomycin treatment, daily dosage, the number of vancomycin trough measurements, serum concentrations of vancomycin, dose adjustments, and the duration of vancomycin treatment. The laboratory results, including estimated glomerular filtration rate (eGFR), serum creatinine, C-reactive protein (CRP), erythrocyte sedimentation rate (ESR), leukocyte count, and cystatin C, were also collected from electronic medical charts.

### 4.4. Vancomycin Concentration Measurements

Vancomycin requires active monitoring to ensure that the serum concentration is within the target range of 15–20 mg/L [17,18]. According to the protocols at Uppsala University Hospital, the trough value is measured just before the third vancomycin dose, along with the eGFR. This process continues until the vancomycin concentration stabilizes, defined as a trough value within the target range in three consecutive samplings without dose or interval adjustments. According to local guidelines, the trough value can be measured every seven days once stability is achieved; however, in clinical practice, controls are typically performed more frequently due to the fear of serum concentrations being outside the target range [17].

### 4.5. Ethics

This study was conducted in accordance with the Declaration of Helsinki and was primarily approved by the Regional Ethical Review Board in Uppsala, Sweden (Dnr 2016/214, date of approval 15 June 2016), and additional approval to access medical charts of the patients in this defined cohort was granted by the Swedish Ethical Review Authority (Dnr 2024-02581-02, date of approval 26 April 2024).

### 4.6. Statistics

The data files were securely stored within the Uppsala Academic Hospital intranet, further protected by a password, and then imported into R (version 4. 3.2 (2023-10-31)) and RStudio (version 2023-09.1+494) for statistical analyses. Quantitative data were summarized using tables, boxplots, and bar charts. Spearman’s rank correlation was employed to assess the correlation between the trough values of vancomycin outside the target range and BMI or age. The same method was applied to evaluate the correlation between the first measured trough value of vancomycin and serum creatinine. Additionally, the Wilcoxon rank-sum test was used to determine if there were differences between sexes or between the patients who received ALBC and those who did not, with respect to the trough values of vancomycin outside the target range. Non-parametrical methods were selected due to the non-normal distribution of most of the investigated data. *p*-values < 0.05 were considered statistically significant.

## 5. Conclusions

A high proportion of vancomycin trough concentrations in PJI patients were outside the therapeutic range, with only slightly more than half falling within the target. Age and pre-treatment creatinine levels were positively correlated with the increasing trough concentrations. Given the frequent need for dose adjustments and the extended treatment duration, alternative agents like dalbavancin should be considered to increase efficiency and to reduce hospitalization.

## Figures and Tables

**Figure 1 antibiotics-13-01226-f001:**
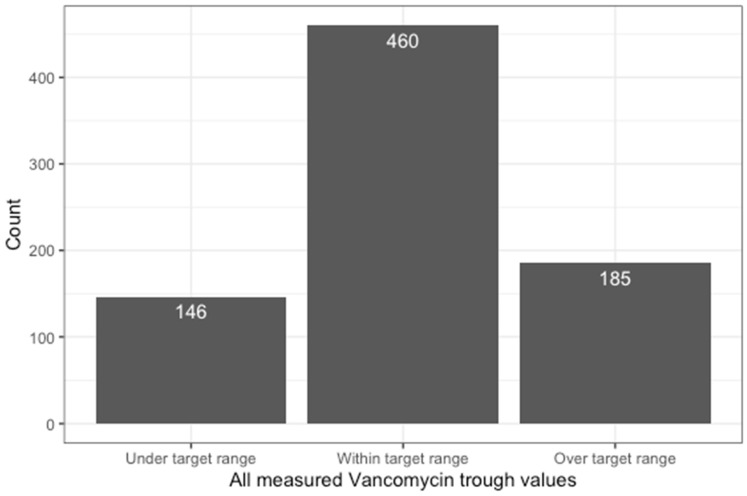
Summary of all measured trough concentrations of vancomycin from all patients.

**Figure 2 antibiotics-13-01226-f002:**
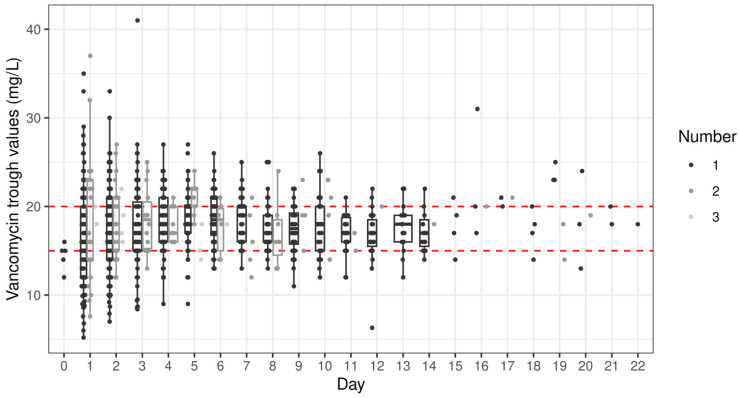
All measured trough values of vancomycin (mg/L) per day after the index procedure. On some days (e.g., day 1), multiple measurements were obtained in some of the patients, indicated by numbers 1, 2 and 3 in the legend, and by different shades of gray in the graph. The red lines indicate the target range, points outside whiskers indicate outliers.

**Figure 3 antibiotics-13-01226-f003:**
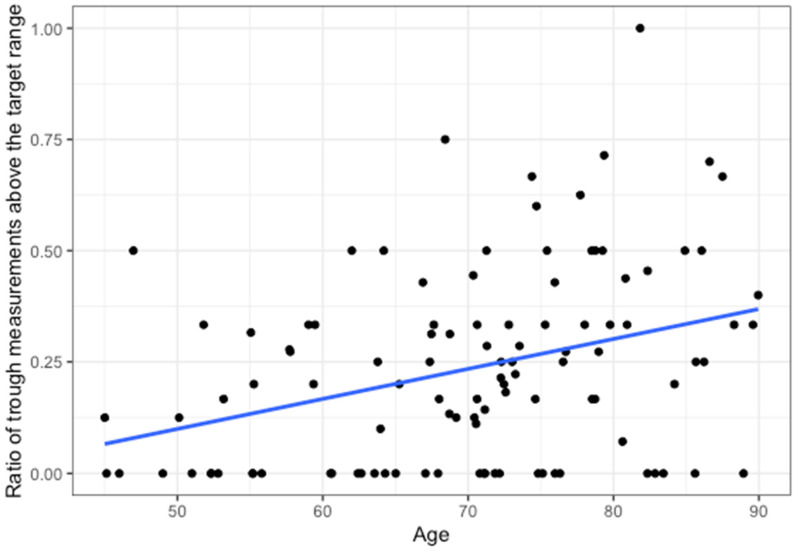
Correlation between the ratio of the number of vancomycin trough measurements above the target range and age (Rho = 0.35, *p* < 0.001). The blue line represents the linear fit.

**Figure 4 antibiotics-13-01226-f004:**
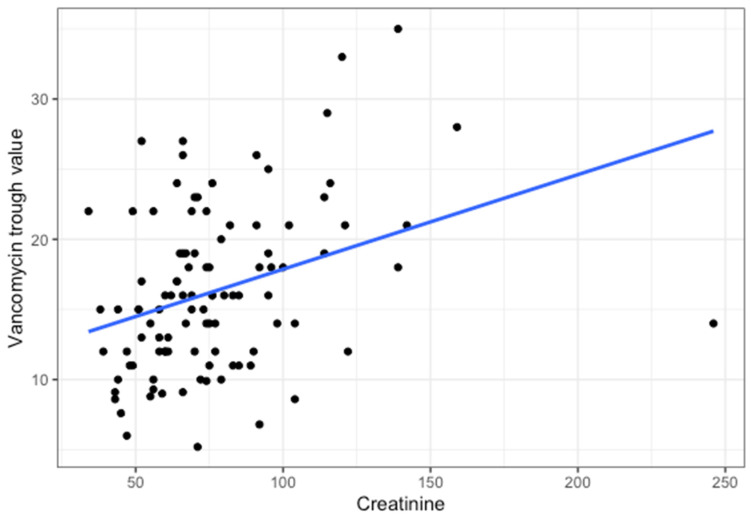
Correlation between the first measured vancomycin trough value (µmol/L) and the creatinine concentration (µmol/L) from the same day (Rho = 0.34, *p* < 0.001). The blue line represents the linear fit.

**Figure 5 antibiotics-13-01226-f005:**
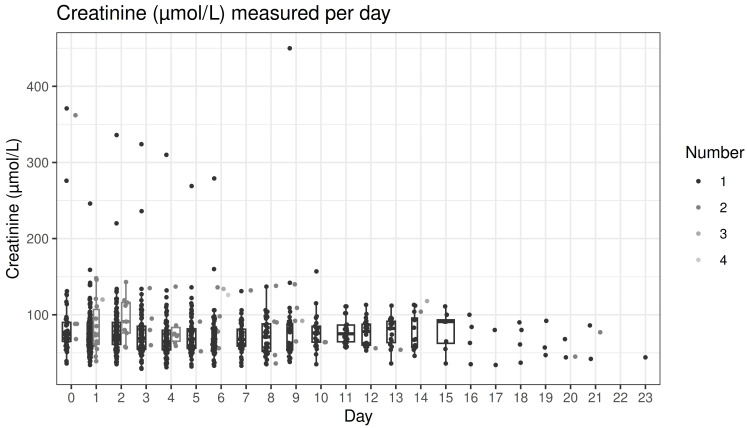
All measured creatinine concentrations (µmol/L) per day after the index procedure. On some days (e.g., day 1), multiple measurements were obtained in some of the patients, indicated by numbers 1, 2, 3, and 4 in the legend, and by shades of gray in the graph.

**Figure 6 antibiotics-13-01226-f006:**
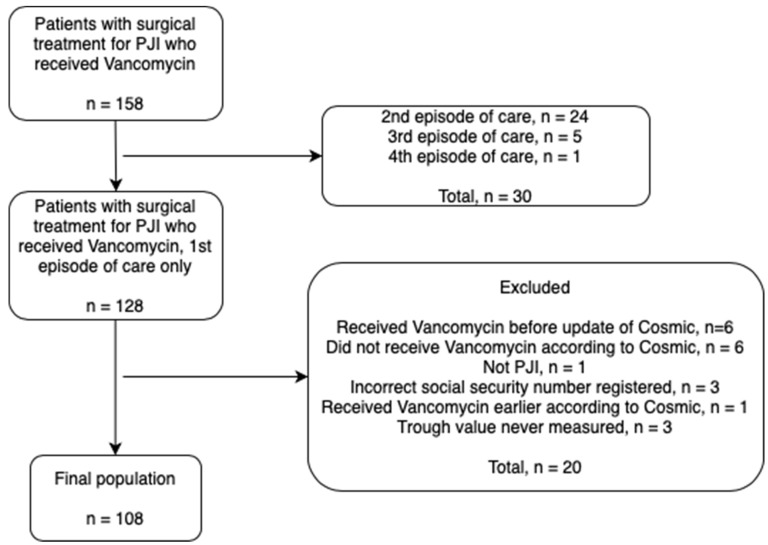
Flowchart of the study population. Cosmic is the electronic medical record system used at our unit.

**Table 1 antibiotics-13-01226-t001:** Description of the study population. ^1^ Median (IQR); n (%).

Study Population	n = 108
Age ^1^	71 (63, 79)
Sex	
- Females	44 (41%)
- Males	64 (59%)
BMI ^1^	27.4 (25.1, 31.1)
- Not registered	7
Joint	
- Hip	79 (73%)
- Knee	29 (27%)
Type of surgery preceding vancomycin administration	
- Other surgery	2 (1.9%)
- DAIR	58 (54%)
- One-stage exchange	19 (18%)
- Two stage exchange part 1	24 (22%)
- Two stage exchange part 2	3 (2.8%)
- Open biopsy	1 (0.9%)
- Not registered	1
Bone cement	
- Vancomycin-loaded	42 (39%)
- No bone cement	66 (61%)

**Table 2 antibiotics-13-01226-t002:** Number of dose changes for the patients.

Number of Dose Changes	n (%)
0	31 (29%)
1	44 (41%)
2	22 (20%)
3	5 (4.6%)
4	4 (3.7%)
5	2 (1.9%)

## Data Availability

According to the Ethical Review Board decision, no other than aggregated data can be shared. After a supplementary application by the responsible researcher, the Ethical Review Board may extend access to the dataset.

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
