# Peer review of "Treatment of Periprosthetic Joint Infection with Intravenous Vancomycin: Do We Hit the Target?"

_antibiotics, 2024, doi:10.3390/antibiotics13121226_

Round 1

Reviewer 1 Report

Comments and Suggestions for Authors

Congratulations on this manuscript. This is an excellent description of current practice in the use of Vancomycin for PJI patients. It has strong application to clinical practice. The authors also recognise the limitations of the research and present these openly.

I would recommend including further detail of the Antibiotic Loaded Bone Cement used in your institution, including dosage of vancomycin, as this has been shown to impact the amount of antibiotic that is eluted. Has your team also considered that the use of antibiotic cement may increase the number of dose changes due to increased initial variability?

Reviewer 2 Report

Comments and Suggestions for Authors

general comments:

The study reveals the fact that the practice of using troughs for monitoring vancomycin is still used in spite of  the current evidence that AUC dosing is more accurate. It does have merit, however a study that would include a more detailed analysis of outcomes would be much more valuable. It is understandable that infection outcomes could not be analyzed due to the fact that data did not include follow up beyond hospital stay, but there are very few comments on the effects on renal function which could potentially be obtained from data from the 8 days of hospital stay. More detailed characteristics of the organisms isolated would be interesting, as - to my knowledge - the troughs targeted are designed and studied for staphylococcus aureus, not for other organisms.

Factors that influenced the length of stay were also not analyzed, which affects the value of the comments / inference that assume the 8 days of hospitalization were related to the vancomycin use and dose adjustment (the authors suggest using a single different drug based on very limited data, rather than improving on the monitoring).

Also, How many patients were bacteremic? the suggested drug, dalbavancin, would not be appropriate for patients who have not cleared bacteremia for instance.

All that said, the value of the study is in the fact that it shows again that vancomycin troughs are difficult to keep in the desired range, they are unpredictable and dosing requires significant effort and cost. And indeed I could not find another similar paper in PJI patients. It is certainly publishable with changes.

Details and minutia

line 9 [abstract) - the intro should mention this is a retrospective single-center case series study

lines 51 and 55 would replace "exogenous" with "contiguous"

line 57 would use "a combination of" instead of "both"

line 68 replace PCR with molecular study to be more general

line 75 replace "not absorbed" with "poorly absorbed" - some absorption does occur; also add streptococci before enterococci.

Line 81 - giving  some detail on the local guidelines and how they were developed may be useful (are these hospital guidelines, swedish guidelines?)

line 107: I would move methods before results

table 1:

would change "type of preceding surgery" to "type of surgery preceding vancomycin administration"

**only 39% had vancomycin in the cement? please comment on this - was this intentional?

line 125 change vankomycin to vancomycin

line 133: change undertaken to obtained 

lines 143 and 268, 282 (maybe in other places that I have missed: change regime to regimen. 

line 170 change registrations to records

176: change red-neck syndrome to "red-person syndrome"

line 184: add high vanco troughs as a risk for toxicity. the toxicity is directly proportional with the levels. levels of 15-20 are associated with - depending on definitions of AKI - as high as 20% rates of AKI.  (Horey nov 1, 2012 Ann of pharm https://journals.sagepub.com/doi/10.1345/aph.1R158 is just one of the multitude of papers that demonstrated that)

line 200: single center is a limitation, that should be added.

201-210, some comment should be made that the current IDSA guideline on treatment of MRSA recommend AUC dosing.

in addition, you quote reference 22 the old guideline but do not mention the new updated guideline from 2020: https://academic.oup.com/ajhp/article/77/11/835/5810200?login=false#google_vignette

This paper https://pubmed.ncbi.nlm.nih.gov/38845212/ should also be commented on. It is a new machine learning algorithm that improves dosing.

The discussion should include this reference and comments on the appropriateness of the local guidelines should be made, given that - if they continue to include trough monitoring 15-20 they are outdated and should be changed. it is understandable that AUC dosing requires different resources and multiple levels.

line 239 and 241, elsewhere in text : change mikrog to mcg

line 259: the word equal is not appropriate, maybe "comparable" would be better. I would also mention that there are no large randomized controlled studies with this drug and the only one is open label randomized 7:1 and included under 100 patients only. It was not powered to look at noninferiority and certainly cannot be labeled as defining "equal" treatments.

line 269: the cost listed here are total hospitalization costs? Also, for the US and worldwide audience, USD may be an easier to understand cost reference, one could mention it for instance in line 155 as "86,684 SEK or approximately 8,000 USD".

Also in the discussion / limitations along with dalbavancin one has to mention other options (daptomycin for instance does not require level monitoring, dose adjustment is much simpler and there is certainly more data. 

figure 5 - in the flowchart, I would replace "Cosmic" with EMR system or mention the EMR system name in the text, the reader has to infer what "Cosmic" is.

Line 329/330 at the end of conclusion I would mention that AUC monitoring should be employed where adequate resources are available.

Round 2

Reviewer 2 Report

Comments and Suggestions for Authors

Thank you for including my suggestions in your article. 

In my view, the article can be published in the current form.

A single minor correction to my own suggestion (I suggested the current term for "red man syndrome" as "red person syndrome" which in fact is now "vancomycin infusion reaction" to be more politically correct.)